# Co-Inoculation of Rhizobacteria and Biochar Application Improves Growth and Nutrientsin Soybean and Enriches Soil Nutrients and Enzymes

**Dilfuza Jabborova** [1,2,3,*], **Stephan Wirth** [3], **Annapurna Kannepalli** [2],
**Abdujalil Narimanov** [1], **Said Desouky** [4], **Kakhramon Davranov** [5], **R. Z. Sayyed** [6],
**Hesham El Enshasy** [7,8,9], **Roslinda Abd Malek** [7], **Asad Syed** [10] and **Ali H. Bahkali** [10]

1   Laboratory of Medicinal Plants Genetics and Biotechnology, Institute of Genetics and Plant Experimental Biology, Uzbekistan Academy of Sciences, Tashkent Region, Kibray 111208, Uzbekistan; narimanov63@list.ru
2   Division of Microbiology, ICAR-Indian Agricultural Research Institute, Pusa, New Delhi 110012, India; annapurna96@gmail.com
3   Leibniz Centre for Agricultural Landscape Research (ZALF), D-15374 Müncheberg, Germany; swirth@zalf.de
4   Botany and Microbiology Department, Faculty of Science, Al-Azhar University, Cairo 11651, Egypt; dr_saidesouky@yahoo.com
5   Institute of Microbiology, Academy of Sciences of Uzbekistan, Tashkent 100128, Uzbekistan; k-davranov@mail.ru
6   Department of Microbiology, PSGVP Mandal's, Arts, Science & Commerce College, Shahada 425409, Maharashtra, India; sayyedrz@gmail.com
7   Institute of Bioproduct Development (IBD), Universiti Teknologi, Malaysia (UTM), Skudai, Johor Bahru 81310, Malaysia; henshasy@ibd.utm.my (H.E.E.); roslinda@ibd.utm.my (R.A.M.)
8   School of Chemical and Energy Engineering, Faculty of Engineering, Universiti Teknologi, Malaysia (UTM), Skudai, Johor Bahru 81310, Malaysia
9   City of Scientific Research and Technology Application, New Burg Al Arab, Alexandria 21934, Egypt
10  Department of Botany and Microbiology, College of Science, King Saud University, P.O. Box 2455, Riyadh 11451, Saudi Arabia; asadsayyed@gmail.com (A.S.); abahkali@ksu.edu.sa (A.H.B.)
*   Correspondence: dilfuzajabborova@yahoo.com

**Abstract:** Gradual depletion in soil nutrients has affected soil fertility, soil nutrients, and the activities of soil enzymes. The applications of multifarious rhizobacteria can help to overcome these issues, however, the effect of co-inoculation of plant-growth promoting rhizobacteria (PGPR) and biochar on growth andnutrient levelsin soybean and on the level of soil nutrients and enzymes needs in-depth study. The present study aimed to evaluate the effect of co-inoculation of multifarious *Bradyrhizobium japonicum* USDA 110 and *Pseudomonas putida* TSAU1 and different levels (1 and 3%) of biochar on growth parameters and nutrient levelsin soybean and on the level of soil nutrients and enzymes. Effect of co-inoculation of rhizobacteria and biochar (1 and 3%) on the plant growth parameters and soil biochemicals were studied in pot assay experiments under greenhouse conditions. Both produced good amounts of indole-acetic acid; (22 and 16 μg mL$^{-1}$), siderophores (79 and 87%SU), and phosphate solubilization (0.89 and 1.02 99 g mL$^{-1}$). Co-inoculation of *B. japonicum* with *P. putida* and 3% biochar significantly improved the growth and nutrient content ofsoybean and the level of nutrients and enzymes in the soil, thus making the soil more fertile to support crop yield. The results of this research provide the basis of sustainable and chemical-free farming for improved yields and nutrients in soybean and improvement in soil biochemical properties.

**Keywords:** biochar; *Bradyrhizobium japonicum*; *Pseudomonas putida*; plant growth; plant nutrients; soil enzymes; soil nutrients; soybean

## 1. Introduction

The global climate scenario is experiencing a drastic depletion of soil nutrients due to various anthropogenic activities, burning of fossil fuel, and excess use of agrochemicals [1]. Applications of plant-growth promoting rhizobacteria (PGPR) and biochar have been advocated as an effective, cheap, and sustainable approach for the replenishment of crop health, crop nutrients, and soil nutrients and enzymes and for improving and sustaining soil fertility [2]. Furthermore, these amendments have a positive impact on the growth [3], development, and yield of several crops [4,5]. Various reports claimed that the application of plant growth-promoting rhizobacteria (PGPR) and biochar improves plant growth, plant nutrients, and physicochemical properties of soil [6–8]. Moreover, such applications of biochar also keep a check onatmospheric $CO_2$ levels [9] and, thus, contribute todecrease global warming effects [10], while the use of PGPR to increase soil fertility and plant nutrients will help to reduce the doses of agrochemicals in the field [11].

A wide variety of symbiotic bacteria, such as *Rhizobium* sp. and *B. japonicum*, etc., have been reported to promote seed germination, the growth of root and shoot, andthe level of nutrients in soybean and also improve soil biochemical properties [4,5].Rhizobia-legumes symbiosis plays a vital role in increasing crop yields, reducing the use of inorganic nitrogen fertilizers and improving soil fertility [12]. Rhizobial species are commonly used as inoculants in various parts of the world for improving the yield of legumes. Co-inoculation with multifarious *Bradyrhizobium* sp. and *Pseudomonas* sp. improves plant growth, plant, and soil nutrients and enzymes through the production of siderophores [13], phytohormones [14], enzymes [15], exopolysaccharide [16], stress tolerance [17], and phosphate solubilization [18–23], etc. Thus, several studies reported increases in nodules number, nodule weight, nitrogen fixed, plant growth, and yield of legumes due to co-inoculation with plant growth promoting *Bradyrhizobium* sp. and *Pseudomonas* sp. [12–14], while the combination of biochar with PGPR further increases root length, shoot length, nodule per plant, seed number, and yield of crops [5].

The activity of PGPR bioinoculants helps in improving the level of extracellular soil enzymes that facilitates the decomposition of soil organic matter and ensures the availability of nutrients in the soil [15]. Among the soil enzymes, proteases and acid and alkaline phosphomonoesterase are the major enzymes that mediate the hydrolysis of the protein and phosphate (P) into bioavailable amino acids, organic nitrogen, and soluble P [16]. However, the activities of these enzymes are governed by many factors, such as soil properties, soil organic matter level, and the presence of organic compounds [24]. We hypothesized that co-inoculation with *B. japonicum*+*P. putida* and biochar would facilitate the beneficial effects on soybean plant growth, plant nutrients, and soil nutrients and enzymes.

The present study was aimed at evaluating the effects of co-inoculation of multiple plant growth-promoting traits positive in *Bradyrhizobium japonicum* USDA 110 and *Pseudomonas putida* TSAU1 and different levels (1 and 3%) of biochar on seed germination, growth parameters, and nutrient levels in soybean and the level of nutrients and enzymes in soil. The outcome of this study may provide a better way of increasing soil fertility and increasing the growth and yield of soybean. This approach has multiple dimensions; as utilization of biochar is not only a cheaper option but will also help in solving the management issues of biochar, it is expected to minimize the doses of agrochemicals and produce chemical-free food. The consortium effect of PGPR and application of biochar provide excellent benefits to the farmers as theyincur less investment and yield more crop productivity, and this organically grown crop has more demand with a good selling price.

## 2. Materials and Methods

### 2.1. Bacterial Culture, Soybean, and Biochar

*B. japonicum* USDA 110 and *P. putida* TSAU1 strains were collected from the culture collection of the Department of Microbiology and Biotechnology, National University of Uzbekistan, Tashkent, Uzbekistan. Soybean (*Glycine max* L. Merr.) seeds were obtained from Leibniz Centre for Agricultural Landscape Research (ZALF), Müncheberg, Germany.

The maize biochar (MBC) was collected from the Leibniz-Institute for Agriculture Engineering and Bioeconomy (ATB), Potsdam, Germany. Pyrolysis of MBC was carried out at 600 °C for 30 min and the chemical compositions of MBC were analyzed according to the method of Reibe et al. [25].

## 2.2. Screening for the Production of PGP Metabolites

*B. japonicum* USDA 110 and *P. putida* TSAU1 strains were screened for phosphate (P) solubilization on Pikovoskaya's agar and in Pikovoskaya's broth [26] for the production of indole-3-acetic acid (IAA)according to the method of Brick et al. [27], for production and estimation of siderophore according to the method of Patel et al. [28] and Payne [29], and the production and estimation of aminocyclopropane-1-carboxylate deaminase (ACCD) activity according to the method of Penrose and Glick [30]. The ACCD activity was measured as the amount of α-keto-butyrate produced per mg protein per h.

## 2.3. Surface Sterilization, Germination, and Bacterization of Seeds

Soybean seeds were sorted to eliminate broken, small, infected seeds and sterilized with 10% sodium hypochlorite solution for 5 min and washed three times with sterile, distilled water. Seeds were germinated in 85 mm × 15 mm tight-fitting plastic Petri dishes with 5 mL of water. *B. japonicum* USDA 110 and *P. putida* TSAU 1 broth rich in PGP metabolites were used for the inoculation of germinated seeds. Germinated seeds were first placed with sterile forceps into bacterial suspension ($5 \times 10^6$ CFU $g^{-1}$) for 10 min before planting, were air-dried, and then planted in plastic pots containing 400 g sandy loamy soil.

## 2.4. Experimental Design

The effect of rhizobacteria on the growth of soybean was studied in pot experiments in a greenhouse at ZALF, Müncheberg, Germany during July 2015. All the experiments were carried out in a randomized block design (RBD) with three replications. Experimental treatments included un-inoculated control (soil without biochar and soil with two levels of biochar (1 and 3%)), inoculation with *B. japonicum* USDA 110 (soil without biochar and soil with two levels of biochar (1 and 3%)), and co-inoculation with *B. japonicum* USDA 110 and *P. putida* TSAU 1 strains (soil without biochar and soil with two levels of biochar (1 and 3%)). The plants were grown in greenhouse conditions at 24 °C during the day and 16 °C at night for 30 days.

## 2.5. Measurement of Plant Growth Parameters and Plant Nutrients

Plants harvested after 30 days were subjected to the measurement of seed germination rate, root length, shoot length, root dry weight, shoot dry weight, and the number of nodules per plant of soybean. Plant nutrients, such as nitrogen (N), phosphorus (P), potassium (K), magnesium (Mg), sodium (Na), and calcium (Ca) were estimated from crushed plant tissue with an inductively coupled plasma optical emission spectrometer (ICP-OES; iCAP 6300 Duo, Thermo Fischer Scientific Inc., Waltham, MA, USA) via Mehlich-3 extraction [30]. The nitrogen and phosphorus contents of root and shoot were determined from dried powdered biomass. For nitrogen estimation, 1 g of plant biomass was digested with 10 mL concentrated $H_2SO_4$ and 5 g catalyst mixture in the digestion tube. The mixture was allowed to cool and then processed for distillation. The distillate was collected and titrated with $H_2SO_4$ blank (without leaf). Total nitrogen was calculated from the blank and sample titer reading [31]. For the estimation of P content, plant P was extracted with 0.5 N $NaHCO_3$ (pH8.5)and treated with ascorbic acid in an acidic medium [32]. The intensity of blue color produced was measured and the amount of P was calculated from the standard curve of P. For the estimation of potassium content of plant biomass, 25 mL of ammonium acetate solution was added in 5 g of the biomass sample, the content was shaken for 5 min and filtered, and the amount of K from the filtrate was measured [33]. For the estimation of Na, Mg, and Ca, 1 g of plant extract was mixed with 80 mL of 0.5 N HC1 for 5 min at 25 °C followed by measurement of concentrations of these elements in the filtrate [34].

### 2.6. Analysis of Soil Nutrient and Soil Enzymes

The rootsoil (10 g) of experimental pots was air-dried soil, shaken with 100 mL ammonium acetate (0.5 M) for 30 min to effectively displace the available nutrients, and adhered to soil minerals. The soil organic carbon (SOC), nitrogen (N), phosphate (P), and potassium (K) content of soil were determined by the dry combustion method according to the method of Sims [35] and Nelson and Sommers [36] using a CNS analyzer (TruSpec, Leco Corp., St. Joseph, MI, USA). For this purpose, 10 mL of 1 N $K_2Cr_2O_7$ and 20 mL of concentrated $H_2SO_4$was added in 1g soil, mixed thoroughly and diluted with 200 mL of distilled water followed by the addition of 10 mL each of $H_3PO_4$ and sodium fluoride. The resulting solution was used for the elemental analysis. Blank (without soil) served as control. Soil Organic Carbon (SOC) of soil sample was calculated with the help of blank and sample titer reading.

The acid and alkaline phosphomonoesterase activities were assayed according to the method of Tabatabai and Bremner [37].Moist soil (0.5 g) was placed in a 15 mL vial, and 2 mL of modified universal buffer (MUB) (pH 6.5 for the acid phosphatase assay or pH 11 for the alkaline phosphatase assay) and 0.5 mL of p-nitrophenyl phosphate substrate solution (0.05 M) were added to the vial, sequentially. The assay and control batches were replicated 3 times. The concentration of p-nitrophenol (p-NP) produced in the assays of acid and alkaline phosphomonoesterase activities were calculated from a p-NP calibration curve after subtracting the absorbance of the control at 400 nm. Protease activity was assayed according to the method of Ladd and Butler [38]. For this, 0.5 g of soil was weighed into a glass vial, and 2.5 mL of phosphate buffer (0.2 M, pH of 7.0) and 0.5 mL of N-benzoyl-L-arginine amide (BAA) substrate solution (0.03 M) were added. The ammonium released was calculated by relating the measured absorbance at 690 nm.

### 2.7. Statistical Analyses

All the experiments were performed in three replicates and the average of triplicate was considered. Experimental data were analyzed with the StatView Software (SAS Institute, Cary, NC, USA, 1998) using ANOVA. The significance of the effect of treatment was determined by the magnitude of the $p$-value ($p < 0.05 < 0.001$).

## 3. Results

### 3.1. Analysis of Maize Biochar

Analysis of pyrolyzed maize biochar contained (g%) dry weight: 92.85, ash: 18.42, total C: 75.16, N: 1.65, P: 5.26, and K: 31.12 with a pH of 9.89 and electrical conductivity of 3.08.

### 3.2. Screening for the Production of PGP Metabolites

Both the cultures under study produced a wide variety of PGP traits. *B. japonicum* USDA 110 and *P. putida* TSAU1 produced 22 and 16 µg mL$^{-1}$ of IAA, 79 and 87% siderophore, and 0.89 and 1.02 99 g mL$^{-1}$ phosphate solubilization, respectively.

### 3.3. Measurement of Plant Growth Parameters and Plant Nutrients

The effect of rhizobacteria and biochar levels indicated a significant improvement in the seed germination rate and growth of the soybean plant treated with biochar and rhizobacteria over the control plant (without biochar treatment). The addition of different levels of biochar, inoculation of *B. japonicum* USDA 110, and *P. putida* strain TSAU 1 with biochar and without biocharshowed variable increases in the growth parameters. Addition of 3% biochar alone enhanced the seed germination by 15%, root length by 20% (Figure 1a), shoot length by 41% (Figure 1a), root dry weight by 22% (Figure 1b), and shoot dry weight by 13% (Figure 1b), as compared to the control plant (without biochar). Individual addition of *B. japonicum* USDA 110 and *P. putida* strains TSAU 1 with varying levels of biochar (1–3%) and without biochar also promoted the growth of the plant. However, a co-inoculation

with *B. japonicum* USDA 110 and *P. putida* strains TSAU 1 with 3% biochar resulted in significant increasesin seed germination and plant growth attributes. Increases in seed germination by 20%, root length by 76% (Figure 1a), shoot length by 41% (Figure 1a), root dry weight by 56% (Figure 1b), shoot dry weight by 59% (Figure 1b), and number of nodules per plant by 57% (Figure 1c) were recorded over the control plant treated with 3% biochar alone.

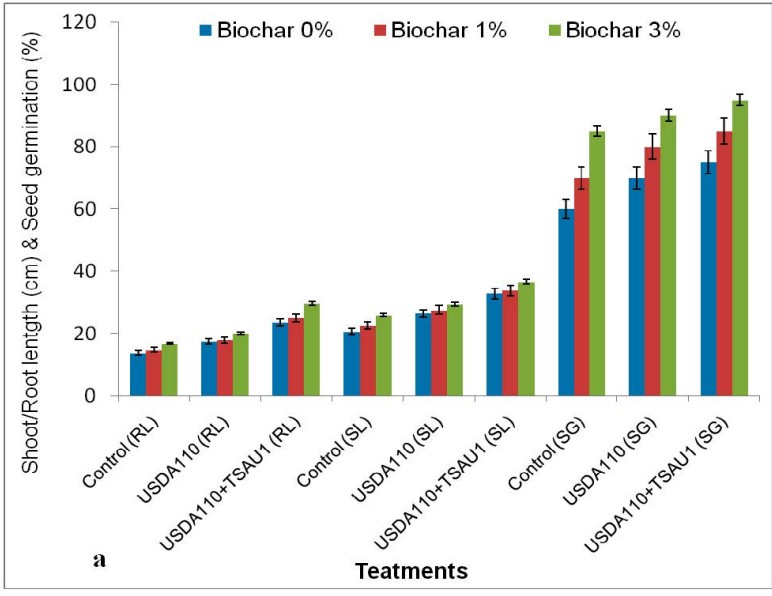

RL= Root length SL= Shoot length, SG= Seed germination *Significant at *P* (0.01)

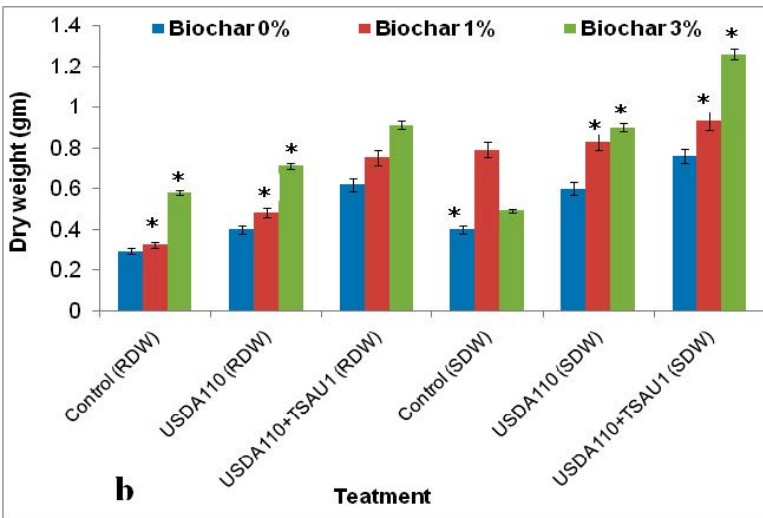

RDW = Root dry weight, SDW = Shoot dry weight, *Significant at *P* (0.01)

**Figure 1.** *Cont.*

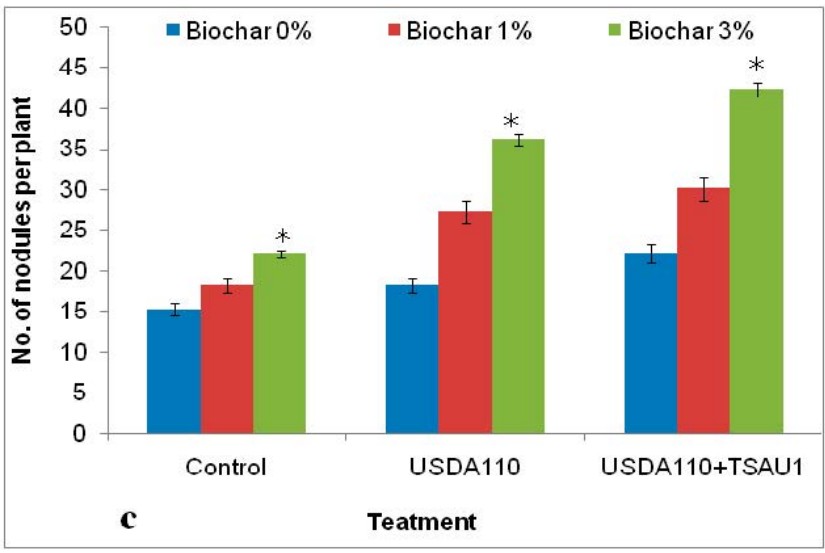

**Figure 1.** Effect of rhizobacteria and biochar concentrations on (**a**) root length [cm] and shoot length [cm], (**b**) dry weight of the root [g] and dry weight of the shoot [g], and (**c**) number of nodules. Plant growth parameters were measured after 30 days of growth of plant growth under greenhouse conditions.* = values significant at *p* 0.01.

Analysis of nutrients in a soybean plant (before sowing and after harvesting) revealedthat treatments with 1 and 3% biochar improved the content of total N, P, K, Mg, Na, and Ca in the plant. The inoculation of *B. japonicum* USDA 110 alone (0% biochar) increased N content by 36%, P content by 8.3%, K content by 5.6%, Mg content by 4.8%, Na content by 30%, and Ca content by 2.88%. However, the co-inoculation of *B. japonicum* USDA 110 and *P.putida* TSAU1 with 3% biochar showed a significant improvement in N content by 62.85%, P content by 7.42, K content by 76.85%, Mg content by 5.14%, Na content by 20%, and Ca content by 28%, as compared to the control (without biochar) (Table 1).

**Table 1.** Effect of rhizobacteria and biochar levels on plant nutrients.

| Biochar Application | Treatments | N (%) | P (%) | K (%) | Mg (%) | Na (%) | Ca (%) |
|---|---|---|---|---|---|---|---|
| 0% | Control | 1.75 + 0.01 | 0.24 + 0.01 | 1.40 + 0.04 | 0.39 + 0.10 | 0.02 + 0.00 | 0.82 + 0.03 |
| | TSAU1 | 2.00 + 0.02 * | 0.25 + 0.04 | 1.41 + 0.02 | 0.43 + 0.02 | 0.06 + 0.01 * | 0.91 + 0.03 |
| | USDA 110 | 2.39 + 0.02 * | 0.26 + 0.04 | 1.49 + 0.02 | 0.47 + 0.02 | 0.08 + 0.01 * | 1.07 + 0.03 |
| | USDA110+TSAU1 | 2.60 + 0.02 * | 0.27 + 0.02 | 2.09 + 0.15 * | 0.62 + 0.01 * | 0.09 + 0.01 * | 1.17 + 0.01 * |
| 1% | Control | 1.77 + 0.02 | 0.27 + 0.03 | 2.33 + 0.02 | 0.66 + 0.02 | 0.03 + 0.01 | 0.95 + 0.03 |
| | USDA 110 | 2.51 + 0.02 * | 0.28 + 0.02 | 2.52 + 0.04 | 0.52 + 0.02 | 0.07 + 0.01 * | 1.25 + 0.03 |
| | USDA+TSAU 1 | 2.64 + 0.02 * | 0.32 + 0.02 | 2.33 + 0.03 | 0.68 + 0.02 | 0.13 + 0.04 * | 1.21 + 0.02 |
| 3% | Control | 1.91 + 0.02 | 0.28 + 0.01 | 2.41 + 0.02 | 0.64 + 0.02 | 0.03 + 0.03 | 1.09 + 0.02 |
| | USDA 110 | 2.27 + 0.01 | 0.37 + 0.01 | 3.64 * + 0.01 | 0.48 + 0.01 | 0.02 + 0.01 | 0.99 + 0.01 |
| | USDA+TSAU1 | 2.85 * + 0.01 | 0.35 + 0.01 * | 3.72 * + 0.01 | 0.39 + 0.01 | 0.03 + 0.01 | 1.04 + 0.01 |

Values are the average of three replicates ± values are standard deviations. Plant nutrient contents were measured after 30 days of growth of plant under greenhouse conditions. * = values significant at *p* 0.01.

### 3.4. Estimation of Soil Nutrient Content and Soil Enzymes

Analysis of soil nutrient content revealed that the inoculation of soybean with *B. japonicum* USDA 110 alone (3% biochar) increased N content by 73%, P content by 173%, and K content by 17%, as compared to the control of 3% biochar. *B. japonicum* USDA 110 alone (3% biochar) significantly enhanced the N content by 98% and K content by 117%, as compared to the control without biochar (Table 2).

**Table 2.** Effect of rhizobacteria and biochar levels on soil nutrients.

| Biochar Application | Treatments | SOC (%) | Total N (%) | P (mg) | K (mg) |
|---|---|---|---|---|---|
| 0% | Control | 21.09 ± 0.01 | 0.080 ± 0.01 | 4.29 ± 0.03 | 2.95 ± 0.02 |
| | TSAU1 | 23.06 ± 0.01 | 0.082 ± 0.01 | 4.43 ± 0.03 | 3.05 ± 0.02 |
| | USDA 110 | 27.08 ± 0.01 | 0.083 ± 0.01 | 4.60 ± 0.02 * | 3.27 ± 0.03 * |
| | USDA+TSAU1 | 29.04 ± 0.02 * | 0.094 ± 0.8 * | 4.88 ± 0.02 * | 5.58 ± 0.03 * |
| 1% | Control | 25.09 ± 0.01 | 0.091 ± 0.01 | 4.22 ± 0.03 | 4.83 ± 0.02 |
| | USDA 110 | 29.06 ± 0.01 | 0.101 ± 0.02 * | 6.14 ± 0.01 * | 5.44 ± 0.01 * |
| | USDA+TSAU1 | 32.07 ± 0.8 * | 0.164 ± 0.03 * | 16.67 ± 0.05 * | 5.68 ± 0.02 * |
| 3% | Control | 25.09 ± 0.01 | 0.094 ± 0.01 | 6.02 ± 0.01 | 5.35 ± 0.03 |
| | USDA 110 | 33.05 ± 0.01 | 0.163 ± 0.01 * | 16.47 ± 0.01 * | 6.30 ± 0.01 * |
| | USDA+TSAU1 | 41.08 ± 0.01 * | 0.170 ± 0.01 * | 18.33 ± 0.01 * | 8.49 ± 0.01 * |

Values are the average of three replicates. ± values are standard deviations. * = values significant at *p* 0.01. Soil nutrient contents were measured after 30 days of growth of plant under greenhouse conditions.

The lowest level of these elements was evident in the soil without biochar treatment. The highest values of SOC, N, P, and K were observed in soil amended with 3% biochar and co-inoculation with *B. japonicum* USDA 110 and *P. putida* TSAU1 vis-à-vis the lowest value found in soil with *B. japonicum* USDA 110 and *P. putida* TSAU1 alone or in combination but without biochar and soil with no bioinoculants and no biochar treatments (Table 2).

Co-inoculation of soybean with of *B. japonicum* USDA 110 and *P. Putida* TSAU 1 strains enhanced nutrient contents of soil compared to all other treatments. The combination with *B. japonicum* USDA 110 and *P. putida* TSAU 1 (3% biochar) significantly increased N content by 80%, P content by 204%, and K content by 58% compared to the control of 3% biochar. When co-inoculated with *B. japonicum* USDA 110 and *P. putida* TSAU 1 (3% biochar)the N content rose by 11% and K content by 35% compared to variants inoculated with *B. japonicum* USDA 110 alone.

The addition of biochar to soil increased the activity of soil protease and acid and alkaline phosphomonoesterase. Substantial increases of 25.05%, 21.02%, and 23.02% in the activities of protease and acid and alkaline phosphomonoesterase, respectively, were evident due to the co-inoculation of *B. japonicum* USDA 110 and *P. Putida* TSAU1 (0% biochar). A combination of this treatment with 1% biochar further improved the activities of these enzymes. However, the activities of these enzymes were significantly improved due to the co-inoculation of *B. japonicum* USDA 110 and *P. Putida* TSAU 1 with 3% biochar. 2-fold, 1.52-fold, and 1.25-fold increases in the activities of protease and acid and alkaline phosphomonoesterase, respectively, were evident due to co-inoculation with two bioinoculants and 3% biochar (Table 3).

**Table 3.** Effect of rhizobacteria and biochar levels on soil enzymes.

| Biochar Application | Treatments | Protease Activity ($\mu g\ NH_4^+$-N $g^{-1}h^{-1}$) | Acid Phosphomonoesterase Activity ($\mu g\ pNPg^{-1}h^{-1}$) | Alkaline Phosphomonoesterase Activity ($\mu g\ pNPg^{-1}r^{-1}$) |
|---|---|---|---|---|
| 0% | Control | 19.2 ± 0.05 | 650.3 ± 30.1 | 300.1 ± 16.3 |
| | TSUA1 | 20.1 ± 0.05 | 697.1 ± 20.1 | 317.1 ± 12.3 |
| | USDA 110 | 23.5 ± 0.10 | 703.3 ± 34.5 | 365.6 ± 18.1 |
| | USDA+TSAU 1 | 25.8 ± 0.19 * | 780.6 ± 38.8 * | 380.2 ± 20.4 * |
| 1% | Control | 21.4 ± 0.07 | 766.3 ± 35.7 | 370.5 ± 19.5 |
| | USDA 110 | 25.8 ± 0.20 * | 820.9 ± 45.3 * | 425.3 ± 21.6 * |
| | USDA+TSAU 1 | 27.7 ± 0.18 * | 940.6 ± 43.2 * | 482.2 ± 20.8 * |
| 3% | Control | 24.3 ± 0.09 | 810.3 ± 37.6 | 420.6 ± 19.5 |
| | USDA 110 | 28.5 ± 0.11 * | 911.8 ± 46.3 * | 483.5 ± 21.2 * |
| | USDA+TSAU 1 | 30.8 ± 0.15 * | 1020.4 ± 48.6 * | 535.7 ± 25.2 * |

Values are the average of three replicates. ± values are standard deviations. * = values significant at $p$ 0.01. Soil enzyme levels were measured after 30 days of inoculation of PGPR and biochar.

## 4. Discussion

### 4.1. Screening for the Production of PGP Metabolites

PGPR is known to produce a wide variety of plant-beneficial metabolites that help in plant nutrition and the overall vigor of the plant [39–42]. Production of IAA, siderophore, and P solubilization have been reported in various species of *Bradyrhizobium*, including *B. japonicum* [42–45] and *P. putida* [46,47]. Sayyed et al. [48] reported the production of siderophores from *P. fluorescence* NCIM5096 isolated from the groundnut field rhizosphere. Shaikh et al. [49] reported the production of siderophore from *P. aeruginosa* isolated from the banana field rhizosphere. Pandya et al. [50] reported the production of siderophore and phytohormones, such as IAA and gibberellins in *Pseudomonas* sp. *Rhizobium* sp., and *Azotobacter* sp. isolated from the sugarcane field rhizosphere. Theyobserved higher yields of phytohormones in *Pseudomonas* sp., as compared to the other isolates. Wani et al. [40] reported the production of siderophore in soil bacterium *P. aeruginosa* RZS9. They claimed a further increase in siderophore yield following the optimization of the process by a statistical approach. Jabborova et al. [14] reported the production of siderophore, IAA, and enzymes, such as protease, cellulose, lipase, P solubilization, and antifungal activity in nine endophytic PGPR strains. Sayyed et al. [51] reported the production of copious amounts of siderophore in *P. fluorescence* NCIM 5096 and *P. putida* NCIM2847.

### 4.2. Measurement of Plant Growth Parameters and Plant Nutrients

An increase in seed germination is due to the phytohormone production, while plant growth promotion during the symbiotic association is due to the nitrogen and other nutrients supplied by the bacterial symbiont. Sayyed et al. [48] reported plant growth-promoting effects of siderophore producing *P. fluorescence* NCIM5096 in wheat and groundnut. Wani et al. [40] reported the plant growth-promoting effects and antifungal-activities production of siderophore producing *P. aeruginosa*. Pandya et al. [50] reported that the inoculation of siderophore and phytohormone producing *Pseudomonas* sp., *Rhizobium* sp., and *Azotobacter* sp. promoted growth in wheat. Jabborova et al. [14] found that inoculation of siderophore, IAA, and enzymes producing P-solubilizing endophytic PGPR strains promoted the growth of medicinal plants. Sayyed et al. [13] observed growth promotion in wheat due to the inoculation of siderophore-producing *P. fluorescence* NCIM 5096 and *P.putida* NCIM2847.

Masciarelli et al. [45] reported a significant increase in the number of root nodules in soybean due to inoculation with *B. japonicum*. Egamberdieva et al. [23] reported the synergistic effect of co-inoculation of *B. japonicum* and *P. putida* to be more effective in increasing nodulation in soybean.Several researchers reported that biochar increased plant growth, nodule number, and yield in different crops [3,5,47]. Pandit et al. [7] claimed that the application of 3% biochar promoted the growth of maize. Uzoma et al. [52] recorded a significant increase in the productivity of biocharrized maize, as compared to a control under sandy soil conditions. Increased growth, more nodulation, and improved yield of soybean after the application of biochar were also reported by Iijima et al. [53].

The addition of organically rich biochar and inoculation with PGPR plays a vital role in increasing the soil microbial activity that provides more nutrition to the plant [54]. Egamberdieva et al. [55] reported significant ($p < 0.05$) increases in N, P, K, and Mg contents in chickpea plants treated with *Mesorhizobium ciceri* and biochar. It has been reported that the biochar amendment improves the water-holding capacity of soil [56], which increases the availability of minerals and nutrients [55]. Shen et al. [57] reported the positive effect of biochar amendment on the plant uptake of plant nutrients. Prendergast et al. [58] claimed that the addition of biochar can induce changes in nutrient availability and may provide additional N, P, K, Mg, Na, Ca. Shen et al. [57] observed an increase in P uptake in plants due to the application of biochar. Egamberdieva et al. [55] observed a significant increase in K content in chickpea roots and shoots treated with *M. ciceri* and biochar. Wang et al. [59] observed similar results and claimed an increasing level of K and Mg uptake in soybean due to the addition of bamboo biochar. Ma et al. [60] reported a positive effect of co-inoculation of *B. japonicum* and biochar on N and other nutrient contents in soybean root and shoot biomass. An increase in N content may be

due to the positive impact of biochar on the nodule number that contributes more N to the shoot and root biomass.

*4.3. Estimation of Soil Nutrients and Soil Enzymes*

Since biochar is an organically rich amendment, its addition is expected to increase the level of soil nutrients. Egamberdievaet al. [55] reported a two-fold rise in SOC, N, P, K, and Mg concentrations in soil amended with biochar, and a three-fold increase in these nutrients in the soil treated with biochar and inoculation with *M. ciceri*. Similar results were reported by Wang et al. [61]. An increase in the soil's organic carbon and other nutrients can also be correlated with increased mineralization due to increased enzyme activity. A linear relationship between soil nutrients and the activities of soil enzymes involved in mineralization has been proposed by Ouyang et al. [62]. Fall et al. [63] reported significant ($p < 0.05$) increases in SOC, available N, soluble P, and total nitrogen upon the application of biochar at a higher rate (12 t ha$^{-1}$). They also recorded an increase in rice rhizospheric carboxylate secretions. Głodowska et al. [6] suggested a combination of biochar and *B. japonicum* strain 532 C, which significantly increased the number of nodules and the growth of soybean. The combination with biochar and *B. japonicum* resulted in enhanced nodulation, nodule biomass, and shoot biomass of soybean [63]. Numerous studies have shown that biocharapplication increases the nutrient contents of plants and soil and improves soil fertility [7,62–64]. Egamberdieva et al. [55] found that inoculation of *B. japonicum* USDA 110 halophilic *P. putida* TSAU1 promoted growth, protein content, nitrogen, and phosphorus uptake and improved the root-system architecture of soybean. Their results indicated that the synergistic effect of co-inoculation of these two strains significantly improved plant growth, nitrogen, phosphorus contents, and contents of soluble leaf proteins as compared with the inoculation with *B. japonicum* USDA 110 alone or the control.

Masciarelli et al. [45] found that co-inoculation of soybean plants with *B. Amyloliquefaciens* subsp. Plantarum and *B. japonicum* showed significant improvement in plant growth parameters and nodulation. They found that inoculation of *B. amyloliquefaciens* subsp. Plantarum with *B. japonicum* enhanced the ability of *B.japonicum* to colonize host plant roots and increase the number of nodules. Phosphomonoesterase (E.C. 3.1.3.2) in the soil is either of plant-root or microbial origin. It plays a major role in P solubilization in soils and in making P available to plants [40]. Acid phosphomonoesterase is dominant in acidic soil, while alkaline phosphomonoesterase occurs in the alkaline soil. The presence of these enzymes and their level in the soil is directly related to the extent of P solubilization and, hence, the amount of soluble P in the soil. Non-nitrogen fixers, such as *Pseudomonas* sp. assimilate nitrogen through the decomposition of protein–nitrogen to low molecular nitrogenous compounds and increase the soil nitrogen and, thus, soil fertility. Extracellular proteases enter the soil via microbial production.

Co-inoculation of *B. japonicum* and *P.putida* along with the application of biochar has been reported to enhance the activities of a wide variety of enzymes in soil [60]. The increase in activities of soil enzyme may be due to increased microbial activity as a result of the addition of consortium of organisms and the addition of biochar that contains good amounts of carbon, nitrogen, and minerals to support cell proliferation and, therefore, enzyme activities [60]. Egamberdieva et al. [55] demonstrated a 2-fold increase in protease and a 40% increase in acid phosphomonoesterase activity due to the addition of biochar. The positive effect on the activities of the soil enzymes can be attributed to the stimulating effect of biochar on microbial activity [63]. The enhancement in the soil enzyme activities due to rhizobial inoculation was also observed by Fall et al. [63]. Ouyang et al. [62] reported that the addition of biochar increases the activities of soil enzymes and attributed this increased enzyme activity to the availability of nutrients and increased microbial activities brought by the addition of biochar to the soil. Egamberdieva et al. [55] and Ma et al. [60] also reported the positive effect of increasing the level of biochar on protease activity. Oladele [64] reported a significant ($p < 0.05$) increase in soil enzymes, such as invertase, alkaline phosphatase, urease, and catalase as a result of the higher application of biochar. It has been reported that with the amendment of more biochar, more soil proteins adhere to the surfaces of biochar pores, make the protein (substrate) unavailable in the soil, and cause a decrease

in protease activity [22]. However, we report increased protease activity with an increase in the biochar amendment to the soil.

## 5. Conclusions

The application of biochar positively affects the growth and nodulation of soybean by increasing nutrient contents, such as N, P, and K in soil. Inoculation with *B. japonicum* USDA 110 alone increased the number of nodules, the length and dry weight of roots, and the length of shoots of soybean, as compared to the control. *B. japonicum* enhanced the total N content, P content, and K content of the soil, as compared to controls with biochar and without biochar, respectively. Co-inoculation with *B. japonicum* USDA 110 and *P. putida* TSAU 1 significantly increased the growth of soybean, nutrient contents in soybeanand soil, and activities of soil protease and acid and alkaline monophosphoeserase, as compared to the control. However, the combined application of *B. japonicum* USDA 110 and *P. putida* TSAU 1 and biochar (3%) showed pronounced positive effects on growth and vigor of soybean, nutrient levels in plant biomass and soil, and activities of soil enzymes. Thus, the co-inoculation with rhizobia and application of biochar offers the best eco-friendly and chemical-freestrategy for the sustainable increase in the yield and replenishment of nutrients in soybean and soil and increase in soil biochemical properties. In general, consortia of PGPR and biochar application improves plant growth, contents of plant and soil nutrients, and soil enzyme activities, which influence soil nutrient retention, nutrient availability, and improve crop growth.The present study demonstrates that application of *B.japonicum* alone has the capacity to improve soybean growth, nutrient contents, and improve soil biochemical properties, however, the co-inoculation of this symbiont along with *P.putida* has a more positive effect on plant growth and soil biochemicals, and co-inoculation of these rhizobia in combination with biochar possesses the capacity to significantly improve the growth and nutrient contents in soybean as well as nutrients and enzyme activities in soil. However, to claim the bio-efficacy potential of the co-inoculation of rhizobacteria and application of biochar needs multiple field studies over the season and in different agro-climatic zones.

**Author Contributions:** D.J. designed the experiments and wrote the manuscript; S.W. and K.D. supervised the work; A.N. and S.D. performed the methodology; R.Z.S., S.W., H.E.E., and A.S. interpreted the data and edited the paper; A.K. and R.A.M. performed analysis, and A.H.B. conceived the funding. All authors have read and approved the paper. All authors have read and agreed to the published version of the manuscript.

**Funding:** This work was funded by The Researchers Supporting Project Number RSP-2020/15, King Saud University, Riyadh, Saudi Arabia, Allcosmos Industries Sdn. Bhd research project No.R.J130000.7344.4B200, and German Academic Exchange Service DAAD 2019, 57440916 for funding this research and UTM-TNCPI research fund for the payment of APC.

**Acknowledgments:** The authors extend their appreciation to The Researchers Supporting Project Number (RSP-2020/15), King Saud University, Riyadh, Saudi Arabia. We thank colleagues at Leibniz Centre for Agricultural Landscape Research (ZALF), Müncheberg, Germany for providing necessary support laboratory and greenhouse facilities, namely the Experimental Field Station and the Central Laboratory.

**Conflicts of Interest:** The authors declare no conflict of interest.

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
