# Peer review of "Co-Inoculation of Rhizobacteria and Biochar Application Improves Growth and Nutrientsin Soybean and Enriches Soil Nutrients and Enzymes"

_agronomy, doi:10.3390/agronomy10081142_

Round 1

Reviewer 1 Report

I think that the manuscript is interesting and concerns an important aspect of the scientific research.

However, there are a numerous minor corrections that needs to be made.

Abstract. The abstract is too long. It should succinctly present an overview of the report. In addition, according to instructions for authors the abstract should be a total of about 200 words maximum.

Introduction. At the end of introduction, the aim of the present study should be mentioned.

Materials and Methods. The Materials and Methods section provides insufficient information to be able to replicate the results. Sections 2.5 and 2.6 should present brief descriptions of the used methods.

There is no information how plant nutrients such as Mg, Na and Ca  were determined in the experiment.

Please provide the details of the Software „StatView“. Developer, version, year, city, country?

Line 138 – Please to add the information – How many repetitions of analysis were performed?

Results. Lines 153-154 – „Addition of 3% biochar alone enhanced the seed germination by 15%“ and Lines 157-159 – „However, a  co-inoculation with B. japonicum USDA 110 and P. putida strains TSAU 1 with 3% biochar resulted in a significant increase seed germination..“. In which figure/table seed germination data are provided?

 There are unnecessary information in the Lines 165-168, 180-184, 193-197 and 221-225. Please delete it.

 The significance of the effect of treatment was determined by the magnitude of the F value. Can you include the F- values of ANOVA in figures and tables?

Lines 163-164 – „Figure 1. Effect of rhizobacteria and biochar concentrations on (A) root length (cm), (B) shoot length  (cm), (C) dry weight of the root (g), (D) dry weight of the shoot (g) and (E) number of nodules“. What the capital letters (A, B, C, D and E) mean in the Figure 1 title?

Secsion 3.3. Measurement of plant growth parameters and plant nutrients. Line 170-171 „Analysis of nutrients in a soybean plant (before sowing and after harvesting) revealed that 1 and 3% biocharrised treatment improves SOC....“ Is it plant nutrients?

References. The references should be corrected according to Instructions for Authors of the journal “Agronomy”.

Author Response

Sl No.

Comments

Response

1

Abstract. The abstract is too long. It should succinctly present an overview of the report. In addition, according to instructions for authors the abstract should be a total of about 200 words maximum.

Revised to present an overview of the paper.

It is of 215words now

2

Introduction. At the end of introduction, the aim of the present study should be mentioned.

Aim of the study is mentioned in the last para of Introduction

3

Materials and Methods. The Materials and Methods section provides insufficient information to be able to replicate the results.

This part is now given in sufficient details

4

Sections 2.5 and 2.6 should present brief descriptions of the used methods.

Sections 2.5 and 2.6 is now written with brief descriptions of the methods used.

5

There is no information how plant nutrients such as Mg, Na and Ca were determined in the experiment.

Information on the estimation of plant Mg, Na, and Ca is now mentioned in section 2.5.

6

Please provide the details of the software “StatView“. Developer, version, year, city, country?

Details of StatView“ software are now mentioned in section 2.7.

7

Line 138 – Please to add the  information – How many repetitions of analysis were performed?

Added

8

Results. Lines 153-154 – „Addition of 3% biochar alone enhanced the seed germination by 15%“ and Lines 157-159. However, a co-inoculation with B. japonicum USDA 110 and Putida /unless otherwise stated putida strains TSAU 1 with 3% biochar resulted in a significant increase seed germination..“. In which figure/table seed germination data are provided?

Seed germination data is now mentioned in Fig 1a

9

There are unnecessary information in the Lines 165-168, 180-184, 193-197 and 221-225. Please delete it.

These unnecessary lines have been deleted now

10

The significance of the effect of treatment was determined by the magnitude of the F value. Can you include the F- values of ANOVA in figures and tables?

The significance of the effect of treatment was determined by the magnitude of the P value and not F value. P values already included in the Tables

11

Lines 163-164. Figure 1. Effect of rhizobacteria and biochar concentrations on (A) root length (cm), (B) shoot length (cm), (C) dry weight of the root (g), (D) dry weight of the shoot (g) and (E) number of nodules“. What the capital letters (A, B, C, D and E) mean in the Figure 1 title?

These capital letters (A,B,C,D,E) are now written as small letter (a,b,c).

(a) Root and shoot length

(b) Root and shoot dry weight

(c) No of nodules

12

Section 3.3. Measurement of plant growth parameters and plantnutrients. Line 170-171 „Analysis of nutrients in a soybean plant (before sowing and after harvesting) revealed that 1 and 3% biocharrised treatment improves SOC....“ Is it plant nutrients?

The word SOC is now deleted from Line No. 170

13

References. The references should be corrected according to Instructions for Authors of the journal “Agronomy”.

References are now formatted as per agronomy

Reviewer 2 Report

The authors conducted a factorial experiments that look at the impact of different combinations of biochar and inoculation of rhizobacteria on the growth of soybean.

The writing is ok but need some structural changes. Especially the discussion need to be improved. Photos of the experiments will be useful. Please see the following detail comments.

Introduction: I think it is a bit short. Maybe you can write a short paragraph about how this study could benefit farmers and the implication to food security.

Section 2.4 Maybe you can draw a table and explain the experiment combination more clearly.

Have you considered doing a set of experiment with just using TSAU 1 and not USDA110? What would you expect?

Line 119. What is the exact date of the experiment? It will be important to know the day length as well. It will be interesting to let the plant grow longer and see if it affects the yield and harvest.

Line 163. I think the figure legend is wrong. Figure 1a is root and shoot length. And there are some figures missing.

Line 180-184. This should be in the Methods. Same for line 193 to 197, 221 to 225. Do not repeat writing the same methods.

Line 214 typo? f ?

Line 230. Write species instead of sp.

Discussion: You should refer the discussion with the results. Compare and contrast with other literature. Some of your Discussion more suit with literature review that should be in the introduction. Also, it will be helpful to add a paragraph to say what is the limitation of your experiment and how can you improve it etc.

Line 264 use three instead of 3

Line 267 does it show linear relationship in your experiment? How do you relate it to your experiment?

I would expect some photographs of the soybean in the experiment. It will be important to show visually how difference with the nodules, shoot length difference.

Author Response

Sl No.

Comments

Response

1

Introduction: I think it is a bit short. Maybe you can write a short paragraph about how this study could benefit farmers and the implication to food security.

The introduction is expanded now

A short paragraph about how this study could benefit farmers and the implication to food security is now added.

2

Section 2.4 May be you can draw a table and explain the experiment combination more clearly.

Experimental combinations are already mentioned in each table. Giving a table for 2.4. will be the repetition.

3

Have you considered doing a set of experiment with just using TSAU 1 and not USDA110? What would you expect?

Yes. a set of experiments with just using TSAU 1 was run and we found a substantial increase in the results. It is now added in all tables

4

Line 119. What is the exact date of the experiment? It will be important to know the day length as well. It will be interesting to let the plant grow longer and see if it affects the yield and harvest.

The exact dates of experiments were 1st July 2015 (sowing) and 30th July 2015.

All these experiments were performed under greenhouse conditions

5

Line 163. I think the figure legend is wrong. Figure 1a is root and shoot length. And there are some figures missing.

Figure 1 legends are now revised

6

Line 180-184. This should be in the Methods. Same for line 193 to 197, 221 to 225. Do not repeat writing the same methods.

This text was written as a footnote for tables. Deleted now

7

Line 214 typo? f ?

Corrected as of

8

Line 230. Write species instead of sp.

Sp. is now written as species

9

Discussion: You should refer the discussion with the results. Compare and contrast with other literature.

Discussion is in line with the results and compared with the literature

10

Some of your Discussion more suit with literature review that should be in the introduction.

Agreed Lines 233-234 are now moved to introduction

11

Also, it will be helpful to add a paragraph to say what is the limitation of your experiment and how can you improve it etc.

Agreed. The limitation of the present study and how it can be improved is given at the end of the conclusion

12

Line 264 use three instead of 3

Revised

13

Line 267 does it show linear relationship in your experiment? How do you relate it to your experiment?

This text is deleted now

14

I would expect some photographs of the soybean in the experiment. It will be important to show visually how difference with the nodules, shoot length difference.

The photograph of experiments are not available now